# Investigation of Laser-Induced Graphene (LIG) on a Flexible Substrate and Its Functionalization by Metal Doping for Gas-Sensing Applications

**DOI:** 10.3390/ijms25021172

**Published:** 2024-01-18

**Authors:** Dongwook Kwak, Hyojin Kim, Seunghun Jang, Byoung Gak Kim, Donghwi Cho, Hyunju Chang, Jeong-O Lee

**Affiliations:** 1Advanced Materials Division, Korea Research Institute of Chemical Technology, 141 Gajeongro, Yuseong-gu, Daejeon 34114, Republic of Korea; dkwak@krict.re.kr (D.K.); hjkim26@krict.re.kr (H.K.); bgkim@krict.re.kr (B.G.K.); roy.cho@krict.re.kr (D.C.); 2Data Research Center, Korea Research Institute of Chemical Technology, 141 Gajeongro, Yuseong-gu, Daejeon 34114, Republic of Korea; jang@krict.re.kr (S.J.); hjchang@krict.re.kr (H.C.)

**Keywords:** laser-induced graphene (LIG), metal-doped graphene, flexible gas sensor, graphene surface functionalization, gas sensing mechanism of graphene

## Abstract

Graphene materials synthesized using direct laser writing (laser-induced graphene; LIG) make favorable sensor materials because of their large surface area, ease of fabrication, and cost-effectiveness. In particular, LIG decorated with metal nanoparticles (NPs) has been used in various sensors, including chemical sensors and electronic and electrochemical biosensors. However, the effect of metal decoration on LIG sensors remains controversial; hypotheses based on computational simulations do not always match the experimental results, and even the experimental results reported by different researchers have not been consistent. In the present study, we explored the effects of metal decorations on LIG gas sensors, with NO_2_ and NH_3_ gases as the representative oxidizing and reducing agents, respectively. To eliminate the unwanted side effects arising from metal salt residues, metal NPs were directly deposited via vacuum evaporation. Although the gas sensitivities of the sensors deteriorate upon metal decoration irrespective of the metal work function, in the case of NO_2_ gas, they improve upon metal decoration in the case of NH_3_ exposure. A careful investigation of the chemical structure and morphology of the metal NPs in the LIG sensors shows that the spontaneous oxidation of metal NPs with a low work function changes the behavior of the LIG gas sensors and that the sensors’ behaviors under NO_2_ and NH_3_ gases follow different principles.

## 1. Introduction

The outstanding features of graphene, which include good thermal and chemical stability, high carrier mobility (~15,000 cm^2^·V^−1^·s^−1^), low electrical noise, and a high specific surface area (~2630 m^2^·g^−1^), have caused it to be actively studied and employed in a wide range of applications [1,2,3,4]. Graphene is composed of a single atomic layer of sp^2^-hybridized carbon atoms packed in a honeycomb crystal lattice in which the carbon–carbon bond length is 1.42 Å. This two-dimensional crystal nature makes graphene an attractive gas-sensing medium, with graphene offering the largest gas-adsorption area per unit volume among the reported gas-sensing media [5]. However, graphene suffers from an inherent lack of selectivity and sluggish gas adsorption and desorption, resulting in slow response and recovery characteristics [6]. Consequently, extensive efforts have been devoted to enhancing the sensing performance of graphene-based gas detectors. To this end, several strategies have been proposed and investigated, including (1) functionalizing graphene with polymers, metal oxides, and noble metals, (2) engineering three-dimensional (3D) graphene nanostructures, and (3) optimizing sensor device configurations [7]. Among these strategies, metal doping into graphene is a well-known, simple, and efficient approach because of graphene’s Dirac cone band structure with zero bandgap, whereby the Dirac points of the graphene reposition its location above or below the Fermi level by electron withdrawal or donating dopants, respectively. Accordingly, this property facilitates the facile modification of its Fermi level via external factors such as doping, leading to improvements in its gas-sensing characteristics [3,8,9]. Numerous first-principles studies based on density functional theory (DFT) have been conducted through measurement of the adsorption energy, bandgap, density of states (DOS), atom–carbon distance, and charge transfer characteristics, among other properties, to reveal how the relationship among metal dopants, graphene, and gaseous analytes affects the sensing response [10,11,12,13,14,15,16,17]; however, the calculations were performed using ideal hypothetical models and not all experimental results can be successfully interpreted with the model system. Cho et al. [7,18] and Zhao et al. found that their experimental results sometimes differed from the theoretical results reported in the literature. They also emphasized that the response process of the metal-doped graphene toward gas exposure is not straightforward; it is complicated by interactions between the metal dopants and graphene, graphene and gas molecules, and even metal dopants and gas molecules. Moreover, few practical and experimental studies on gas detection using metal-doped graphene have been reported, especially concerning graphene with 3D structures and defects, to support the theoretical studies. To develop a high-performance graphene-based gas sensor with metal-doped graphene, clarifying the sensing mechanisms through correlated theoretical and experimental studies is strongly advised.

Compared with conventional techniques for the synthesis of graphene (e.g., chemical vapor deposition (CVD); micromechanical, electrochemical, and chemical exfoliation of graphite; epitaxial growth), the laser direct writing (LDW) method provides distinct benefits such as rapid production, precision patterning capability, and the ability to fabricate graphene under ambient conditions, even though the obtained graphene is multilayered and contains a certain concentration of defects [2,19,20]. In particular, due to the capability of rapid production and precision patterning, a considerable cost reduction in manufacturing and high-resolution fabrication can be achieved, leading to advances in nanostructured graphene. In addition, the LDW route is not only a scalable and effective method to manufacture 3D graphene with a structure that is more favorable for gas-sensing applications than the structure of 2D graphene but also enables the functionalization of the graphene surfaces. Both top-down and bottom-up syntheses of 2D graphene involve high temperatures or harsh chemicals, and the irregular stacking of the materials hinders the functionalization of surfaces [21]. An exceptional advantage of LDW over traditional methods is that it enables different physical properties (e.g., superhydrophilicity or superhydrophobicity) and various microstructures (e.g., sheet, foam, fiber, and nanodiamond formation) to be obtained by simply manipulating atmospheric conditions and controlling the laser’s parameters, respectively [22]. To realize such benefits, laser-induced graphene (LIG) has been actively investigated and is used in sensor applications such as biosensors, humidity sensors, strain sensors, temperature sensors, piezo-resistive sensors, wearable body-condition sensors, and gas sensors [1,2,23]. In general, LIG can be formed on various substrate materials (e.g., polyetherimide (PEI), poly(ether ether ketone), polyethersulfone, polyimide (PI), polysulfone, paper, wood, and potato); however, in the case of gas-sensing applications, PI has been most frequently selected because of its good radiation resistance, low noise generation, exceptional thermal stability, and good endurance in harsh environments [24,25].

In this article, we experimentally demonstrate and assess the gas-sensing responses of LIG-based gas sensor devices that have been synthesized on a flexible PI substrate for sensing NO_2_ and NH_3_, which are used as representative oxidizing and reducing agents, respectively. Their sensing mechanisms and the influence of various metal dopants (Ag, Al, Au, Cu, In, and Pd) on the sensor response to the gaseous analytes are also elucidated. Because our LIG exhibits p-type semiconducting behavior, both our pristine LIG (P-LIG) and metal-doped LIG (M-LIG) sensor devices exhibit negative responses (decreasing resistance) to an oxidizing agent (NO_2_) and positive responses (increasing resistance) to a reducing agent (NH_3_); however, the interactions among each type of metal nanoparticle (NP), LIG, and gas molecule alters the devices’ electronic properties, which determines whether their sensing performance is enhanced or deteriorated. This work might provide guidance in selecting an appropriate metal dopant for 3D-structured graphene with a certain degree of defects for use in specific gas detection, enabling the development of a flexible and high-performance LIG gas sensor. Also, since this work focuses on the electronic structures of the graphene and dopant, it could contribute to the enhancement of other decorated 3D carbon-based applications, such as supercapacitors, fuel cells, batteries, photodetectors, etc., which utilize carbon nanotubes (CNT), carbon nanofiber (CNF), carbon foams (CF), fullerene, and so on as their basal carbon source [26,27,28,29].

## 2. Results and Discussion

### 2.1. Fabrication and Characterization of the P-LIG Sensor Device

Figure 1a shows the P-LIG sensor, which has a serpentine pattern in the middle and two circles on the top and bottom (used for the Ag-pasted contact areas), synthesized on a flexible PI substrate. Upon irradiation of the substrate with a 10.6 μm CO_2_ laser under ambient conditions, a pattern with a width of ~400 μm and a thickness of ~22 μm, along with a circle with a 500 μm radius and a thickness of ~22 μm, were formed on the 125 μm-thick PI film (Figure 1b and Appendix A). Figure 1c confirms that the synthesized LIG had a highly porous structure, which is beneficial for gas adsorption because of its remarkably high specific surface area.

The Raman spectrum in Figure 1d shows the three typical peaks of graphene: the D peak (~1350 cm^−1^), the G peak (~1583 cm^−1^), and the 2D peak (~2695 cm^−1^). In general, the D peak is related to the breathing mode of sp^2^ carbon in an aromatic formation, representing the existence of defects and disorder in graphene and serving as an indirect measure of the graphene’s quality. The G and 2D bands correspond to the in-plane vibration mode of sp^2^-hybridized carbon networks and the stacking order of graphene along its *c*-axis, respectively. Because the I_2D_/I_G_ and I_D_/I_G_ values obtained from the Raman spectrum are ~0.69 and ~0.79, respectively, the synthesized P-LIG is, therefore, assumed to have a multilayer structure with a relatively low defect density [30]. In addition, on the basis of the Tuinstra–Koenig relationship, the crystalline size of the P-LIG along the *α*-axis (*L_a_*) was calculated to be ~389.36 nm, using Equation (1) [30]:(1)$$L_{a}=\left(2.4\times 10^{-10}\right)\times \lambda_{l}^{4}\times \left(\frac{I_{G}}{I_{D}}\right)$$
where *λ_l_* is the wavelength of the Raman laser (*λ_l_* = 1064 nm).

The full-survey XPS spectrum in Figure 1e clearly shows that only the C1s (282.3 eV) and O1s (530.7 eV) peaks are detected in the spectrum of P-LIG and that their calculated atomic percentages are 94.9% and 5.1%, respectively, confirming the successful synthesis of the LIG.

### 2.2. Metal-Doping of P-LIG and Characterization of the M-LIG Sensor Devices

In the present study, the vacuum-evaporation method was used to functionalize LIG devices with metal NPs because the generally used method of heteroatom doping by direct writing in salt baths can result in unwanted residues [31]. As shown in Figure 2a, a metal stencil mask made of SUS304 was used to deposit metal NPs onto the prefabricated P-LIG sensor device, and a total of 18 sensor devices were produced through a single deposition via thermal evaporation. A metal thickness of 4 nm was used in the fabrication of all of the M-LIG sensors because the formation of metal NPs was apparent when the metal deposition thickness was 4 nm (as measured using a QCM), as shown in Appendix A. A computer-based design tool (AutoCAD and CorelDRAW) enabled fine alignment between the stencil mask and the P-LIG sensor devices; the EDS mapping images presented in Figure 2b and Appendix A revealed that the deposited metal NPs were well-dispersed over the targeted serpentine area and on the surface of the LIG. To observe the interaction between the metal dopants and the LIG, we carried out XPS analysis. Compared with the surface of the P-LIG, the surfaces of all of the M-LIGs exhibited substantial metal oxidation (Figure 2c). In addition, the high-resolution survey (Appendix A) and quantification results (Appendix A) reveal that greater oxidation occurred in the Al-, Cu-, and Pd-doped LIG devices than in the devices doped with other metals. Notably, the readily oxidized AlO*_x_* shell–Al NPs can be formed on the surface of graphene because the Al metal has a high oxidation rate; thus, the Al-doped LIG sensor device exhibited the highest degree of oxidation among the investigated devices [32].

### 2.3. Gas-Sensing Performance Characterizations

With increasing temperature, our P-LIG sensor device shows a decrease in resistance (Appendix A); its responses toward 1000 ppm of NO_2_ were examined to determine the optimal operating temperature. The relative response, ∆R/R_0_ (%), of the sensor was calculated using Equation (2):(2)$$∆R/R_{0}(\%)=\left(\frac{R_{g}-R_{0}}{R_{0}}\right)\times 100$$
where *R_g_* and *R*_0_ are the sensor resistances under the analyte gas and under N_2_, respectively. Appendix A shows comparison graphs for the response of the P-LIG sensor device to 1000 ppm of NO_2_ as the temperature was varied from 50 °C to 150 °C. The response shows an approximately twofold increase as the temperature was increased from 50 °C to 100 °C; however, once the response reached its maximum at 100 °C, it decreased as the temperature was increased further to 150 °C, where its value was even lower than that at 50 °C. Because adsorption and desorption are temperature-sensitive processes, the deterioration in response at 150 °C might be attributable to a decrease in the Debye length as the charge-carrier density increases at high temperatures and also to the untimely desorption of the adsorbed gas molecules that are bound to the graphene before an electrical interaction can occur between them [33,34,35]. These factors might lead to a shift of the baseline resistance, thereby hindering the detection of the minute variation in resistance that occurs via gas adsorption or desorption. Therefore, a temperature of 100 °C was selected as the optimal operating temperature for all of the gas-sensing tests.

After optimizing the operation temperature, we exposed the P-LIG sensor device to different gas concentrations ranging from 250 to 1500 ppm for both NO_2_ and NH_3_ to investigate the sensitivity of the LIG devices. The results in Appendix A show that the response saturation started from 500 ppm of NO_2_, whereas no substantial variation in response was observed for elevated NH_3_ concentrations. Because the highest responses were detected for 1000 ppm of NO_2_ and also for 1000 ppm of NH_3_, a gas concentration of 1000 ppm was chosen for the NO_2_/NH_3_ response tests to evaluate the influence of metal dopants on the LIG devices’ gas-sensing performance.

Figure 3 shows a set of seven single-response cycles for the P- and M-LIG sensor devices when exposed to 1000 ppm NO_2_ (a) and NH_3_ (b) at 100 °C while a DC bias of 1 V was applied. In fact, all of the tests were run with three on–off cycles and each cycle consisted of 10 min of analyte-gas purging and 25 min of N_2_ purging for the “on” and “off” cycles, respectively (Appendix A). When the devices were exposed to NO_2_, a negative trend in ∆R/R_0_ (%) was observed for all of the devices as the resistance decreased, whereas a positive tendency was observed as the resistance increased upon exposure to NH_3_. This typical phenomenon can be observed when graphene with a p-type conducting characteristic is in contact with an oxidizing agent (i.e., the electron acceptor, NO_2_) or a reducing agent (i.e., the electron donor, NH_3_) [10,36,37]. In addition, graphene synthesized under ambient conditions typically exhibits p-type conduction, where the major carriers are holes, because of adsorbed oxygen or water molecules [38]. Thus, as depicted in Figure 4, once the p-type graphene is exposed to an oxidizing NO_2_ gas, hole accumulation within the graphene occurs after the electron migration from graphene to the gas molecules, which results in a decrease in resistance. Meanwhile, upon the sensors coming into contact with a reducing NH_3_ gas, electron injection from the gas molecules into the graphene reduces the major carrier concentration of the graphene; as a result, the overall resistance increases. In addition, we note that the sensitivity (S,$\left|∆R/R_{0}(\%)\right|$) of both the P- and M-LIG sensor devices toward NO_2_ gas is much higher than that toward NH_3_ gas. As the first-principles study validated, this difference might be attributable to the higher adsorption energy (~67 meV) and charge transfer (~0.099 e from graphene to a gas molecule) of NO_2_ compared with those of NH_3_ (~31 meV and ~0.027 e for the adsorption energy and charge transfer from the molecules to graphene, respectively) when they are adsorbed onto the surface of graphene [10,39].

To investigate the role of metal incorporation into the LIG on the gas-sensing performance of LIG-based sensors, we selected metal dopants that are frequently used in first-principles studies of metal-doped graphene (i.e., Ag, Al, Au, Cu, In, and Pd) to fabricate the M-LIG sensor devices [11,12,13,14,16,17,39,40,41,42,43]. As shown in Figure 5, the NO_2_ response of the P-LIG device was superior to those of all of the M-LIG devices. This deterioration in the NO_2_ response of the M-LIG sensor devices cannot be explained by a simple difference in work function between the metal NPs and the LIG. (When work-function differences between pure metals and graphene are considered, the sensitivity should show a trend of high-work-function metal-doped LIG < LIG < low-work-function metal-doped LIG.) Rather, the deterioration can be explained by considering the sophisticated interactions between the metal NPs and the LIG, the metal NPs and NO_2_, and the LIG and NO_2_. To this end, an interpretation based on the electronic structures of metal NPs, LIG, and gaseous analytes can be applied to understand the inferior responses of the M-LIG sensor devices toward NO_2_.

First, we focused on the oxidation of metals on the surface of the LIG. According to the analyzed XPS data, the M-LIG sensor devices with highly oxidized surfaces (Al, Cu, and Pd) demonstrate weaker responses to NO_2_ than those devices with metals with relatively weakly oxidized surfaces (Ag, Au, and In). In general, the oxidation of a metal dopant on a sensing layer can modify the surface properties and the overall electronic structure of the sensing material by changing the charge state [44,45,46]. Moreover, the oxidized metal dopant can react with a vacancy in the host material (p-type) at a relatively high temperature (~100 °C) to generate electrons via the electronic compensation mechanism. These electrons can contribute to a reduction in the hole carrier concentration through electron–hole recombination, negatively affecting the sensing performance of the p-type LIG [47,48,49]. In addition, compared with the pure metallic catalysts on the LIG, their undesirable oxidation might diminish the spill-over effect by shrinking its active reaction site for gas molecules and, consequently, decreasing the sensitivity [50,51]. Among the various highly oxidized metal-doped LIG devices, the NO_2_ response is ranked in order of Al, Cu, and Pd and their work-function order is Pd > Cu > Al [12]. The slightly stronger response of the Al-doped LIG device might originate from the formation of a hole-depletion zone in the LIG that is adjacent to the interface with Al NPs [52]. Because the work function of Al is lower than that of multilayered graphene, a Schottky contact is generated by a large Fermi-level difference between the Al metal and p-type graphene, which leads to the formation of a hole-depletion zone in the LIG [12,53]. This hole depletion promotes electron charge transfer from the LIG to the gas molecules (Figure 6) and, thus, enhances the sensing response to NO_2_ [7]. In contrast, the adsorption of the higher-work-function Cu onto LIG creates a hole accumulation zone at the interface via the p-doping effect on the LIG [52]. This hole accumulation zone impedes electron charge transfer from the LIG to NO_2_ molecules, thereby weakening the response to NO_2_.

Oddly, the surface of the Pd-doped LIG is highly oxidized, even though Pd is a noble metal; consequently, the Pd-doped LIG device exhibits the weakest response among the investigated devices. In addition to its highly oxidized surface, the weakest response of the Pd-doped LIG device toward NO_2_ might also be attributable to the Pd substantially damaging the conical points of graphene at *K* via hybridization between the graphene *p_z_* states and the Pd *d* states, impeding the charge transfer of graphene [14]. In the case of relatively less-oxidized metals, the M-LIG devices show a response order of Ag, Au, and In toward NO_2_. Although the work function of In is lower than that of Ag and Au, its corresponding response is inferior to the others and is even similar to that of highly oxidized metals [13]. Chandni et al. [42,43]. and Jia et al. investigated the transport of In adatoms on graphene and found that the In adatoms drastically reduced the carrier mobility of the graphene and increased the level of charge-density inhomogeneity in the graphene, possibly resulting in a weaker response toward NO_2_. In a comparison of Ag- and Au-doped LIG devices, the Ag-LIG device exhibited a slightly stronger response because of the lower work function of Ag [12].

In contrast to the NO_2_ response results, all of the investigated metals appear to have participated in the enhancement of the NH_3_ gas response (Figure 3b and Figure 5). An approximately three- to four-fold increase in relative response was observed in the M-LIG sensor devices compared with the P-LIG device. As previously mentioned, NH_3_ exhibits lower adsorption energy and a lower charge transfer rate than NO_2_; in this case, the chemical sensitization effect of metal NPs on the LIG might play a dominant role in the sensing mechanism for NH_3_ detection. In general, metal NPs on a sensing film tend to offer more active sensing sites for the analyte gas and to catalytically promote the dissociation of gas molecules into a more reactive status, leading to sensitivity enhancement [54,55,56]. Similar to the influence of metal oxidation on the NO_2_ response, greater metal oxidation led to a lower NH_3_ response; notably, however, reducing the charge carrier mobility by introducing In as a dopant might profoundly lower the sensitivity, resulting in the weakest response toward NH_3_. Meanwhile, among the M-LIG sensor devices, the Ag-doped LIG device exhibited the strongest response to NH_3_ as its ranked order in response to NO_2_.

## 3. Materials and Methods

### 3.1. Synthesis of LIG on PI

A serpentine-patterned LIG (Figure 7) was fabricated directly onto a commercial PI film (125 μm-thick Kapton polyimide film, DuPont, Wilmington, DE, USA) with a CO_2_ laser system platform (VLS 2.30DT, power *P*_max_ = 30 W, speed *S*_max_ = 1270 mm·s^−1^, and wavelength *λ* = 10.6 μm, Universal Laser Systems, Scottsdale, AZ, USA). The two circle shapes on the LIG were formed on each top and bottom edge to provide electrical connections. The optimized laser parameters were a laser power of 4 W, a scanning speed of 127 mm·s^−1^, and an image density of 500 pulses per inch (PPI, 1 inch = 25.4 mm); these parameters were used for fabricating all of the devices. The desired pattern was designed via AutoCAD and CorelDRAW.

### 3.2. Preparation of P-LIG and M-LIG Devices

As shown in Figure 7, metal NPs (Ag, Al, Au, Cu, In, and Pd) were deposited onto the prefabricated P-LIG device with metal (SUS304) stencil masks, using a thermal evaporator system. For each metal deposition process, a separate mask was used to avoid cross-contamination; in addition, the metals were deposited to a thickness of 4 nm, which was measured using a quartz crystal monitor (QCM), at a deposition rate of ~0.2 Å·s^−1^ under a pressure of ~1 × 10^−7^ torr. After the deposition, the top/bottom circle areas were coated with Ag paste to allow all of the devices to connect to the external electronics.

### 3.3. Bare LIG and Metal-Doped LIG Characterizations

The 3D porous structure of the LIG was characterized via scanning electron microscopy (SEM, Philips XL30S, FEI, Eindhoven, The Netherlands). Energy-dispersive X-ray spectroscopy (EDS, Bruker Quantax 200, Billerica, MA, USA) was performed with an XFlash6 Si-drift detector to observe the dispersed metal NPs on the LIG. X-ray photoelectron spectroscopy (XPS) and Raman spectroscopy were carried out using a photoelectron spectrometer (AXIS Supra+, Kratos Analytical Ltd., Manchester, UK) and a laser (1064 nm Nd:YAG) Raman spectrometer (Bruker FRA 106/S, Billerica, MA, USA), respectively.

### 3.4. Sensor Testing Equipment

A custom-designed testing chamber with a volume of 100 mL and a homemade probe station with a ceramic heater were constructed to conduct the gas-sensing test. Both the gas concentration and the flow rate of the gas analytes were precisely regulated by mass flow controllers (MFCs) that were connected to the inlet of the chamber. At a constant flow rate of 1 L·min^−1^, every test was run with three on/off cycles, whereby an individual cycle consisted of 10 min of purging with the gas/N_2_ mixture, followed by 25 min of purging with pure N_2_. Two probes of the probe station were attached to the Ag-pasted areas of the sensor device and were used to measure the current (*I*) and voltage (*V*) while a constant DC voltage (1 V) was applied to the device. The real-time change in the electrical resistance of the device was monitored and recorded using a source meter (Keithley 2612B, Keithley Instruments, Cleveland, OH, USA). The temperature of the ceramic heater in the probe station was controlled by an electronic heating system, and the operation temperature was optimized at 100 °C (Appendix A) for all of the tests.

## 4. Conclusions

The LDW method, which is known as a facile and effective route for fabricating graphene, was used to produce LIG gas-sensor devices on flexible PI films, and their sensing performance regarding an oxidizing agent (NO_2_) and a reducing agent (NH_3_) was investigated. For comparison, various metals such as Ag, Al, Au, Cu, In, and Pd were used to construct the M-LIG sensor devices; the influence of metal incorporation on the sensing response was also observed. The experimental results showed that our LIG has a certain degree of defects and multilayers and that doping metals into this graphene led to an enhanced response to NH_3_ gas but a diminished response to NO_2_ gas. This result is attributable to the metal NPs providing a large specific surface area for the LIG, enabling the detection of NH_3_ gas molecules, which have a lower adsorption energy and less charge transfer than NO_2_ molecules. Conversely, NO_2_ gas sensing might be strongly affected by the alteration of the electronic structure of the M-LIGs via either the oxidation of metal dopants on the LIG surface or the formation of hole-depletion/accumulation areas at the interface, as a result of the work-function difference between the metal dopant and the LIG. Given the lack of relevant practical research on the gas-sensing performance of metal-doped 3D-structured graphene with defects and multilayers, as in the case of LIG, this work could not only lead to a better understanding of its sensing mechanism through clarifying the interactions between the LIG and gas molecules, metal dopants and the LIG, and gas molecules and metal dopants but could also support first-principles studies of metal-doped graphene systems in gas detection, ultimately contributing to the development of high-performance LIG-based gas sensors. Furthermore, since this study investigated the relationship of the electronic structure between graphene and metal dopants, its results could be exploited for the enhancement of LIG-based supercapacitors, especially wearable energy storage devices, and could also be utilized for the improvement of flexible photodetectors using Schottky junctions, constructed by controlling the work function of LIG and incorporated materials such as metal or metal oxide [31,57].

## Figures and Tables

**Figure 1 ijms-25-01172-f001:**
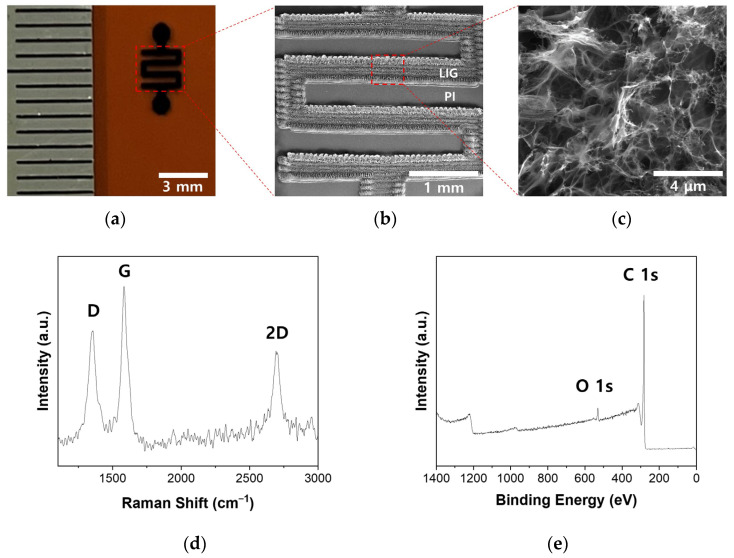
(**a**) Optical image of a P-LIG gas sensor on a PI film. (**b**,**c**) SEM image of the serpentine area with a ~400-μm-thick width. (**d**,**e**) The Raman spectrum and XPS survey spectrum, respectively, of the P-LIG.

**Figure 2 ijms-25-01172-f002:**
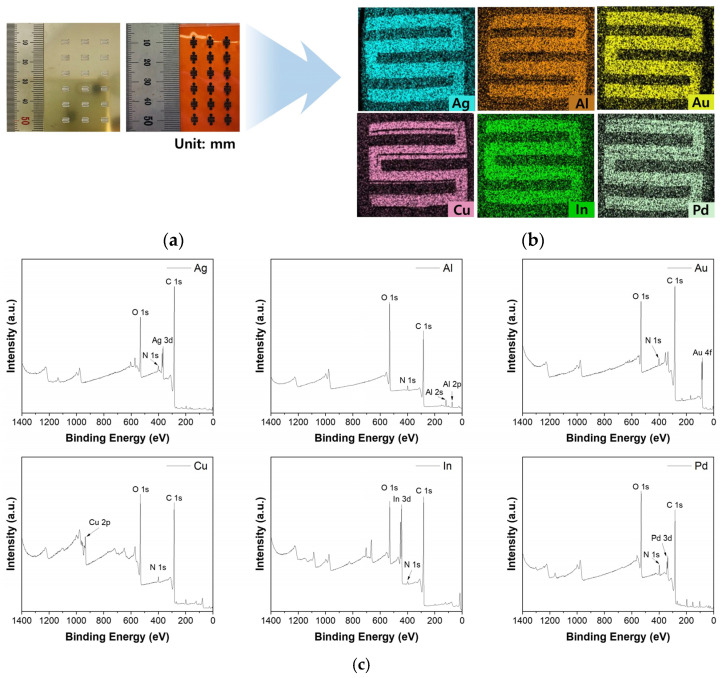
(**a**) Metal stencil mask (left) and the serpentine P-LIG patterns (right). (**b**) EDS mapping images for the M-LIG devices. (**c**) XPS survey spectra of the M-LIG devices.

**Figure 3 ijms-25-01172-f003:**
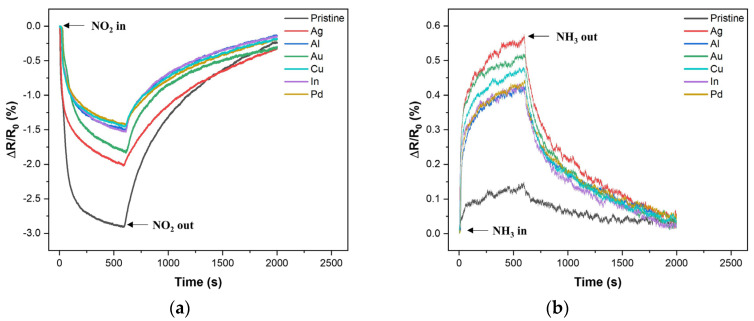
Comparison plots for the responses of the P- and M-LIG sensor devices toward (**a**) 1000 ppm of NO_2_ and (**b**) 1000 ppm of NH_3_.

**Figure 4 ijms-25-01172-f004:**
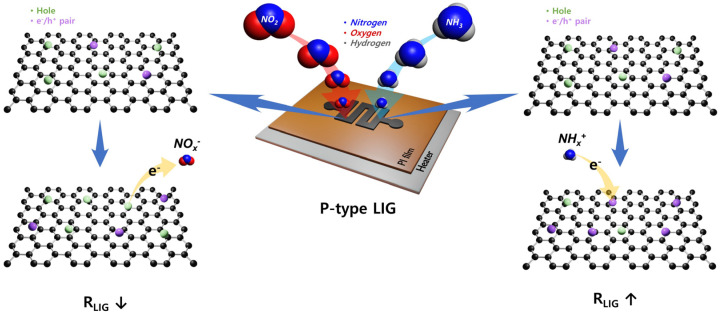
Schematics of the gas-sensing mechanisms of the P-LIG sensor device toward NO_2_ (**left**, oxidizing gas) and NH_3_ (**right**, reducing gas).

**Figure 5 ijms-25-01172-f005:**
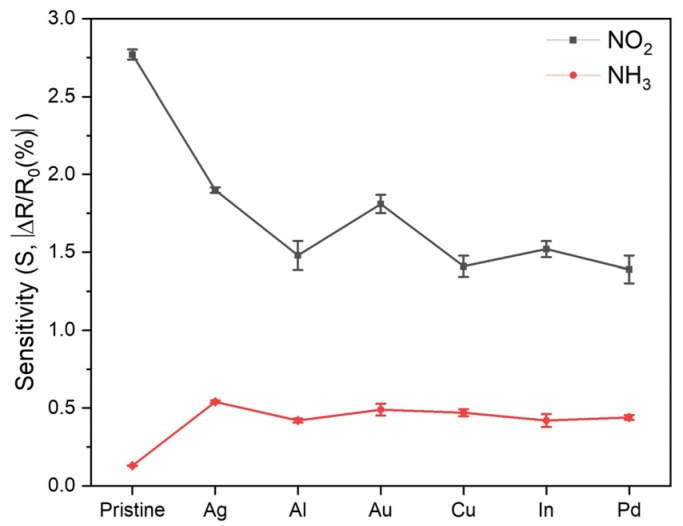
Sensitivity comparison plot for the P- and M-LIG sensor devices upon exposure to NO_2_ (black) and NH_3_ (red).

**Figure 6 ijms-25-01172-f006:**
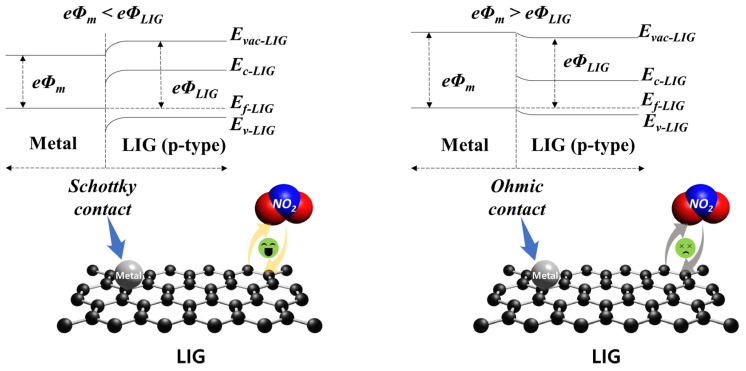
Schematics showing the formation of a Schottky contact (*Φ*_m_ < *Φ*_LIG_, **left**) and an ohmic contact (*Φ*_m_ > *Φ*_LIG_, **right**) in the M-LIG sensor devices.

**Figure 7 ijms-25-01172-f007:**
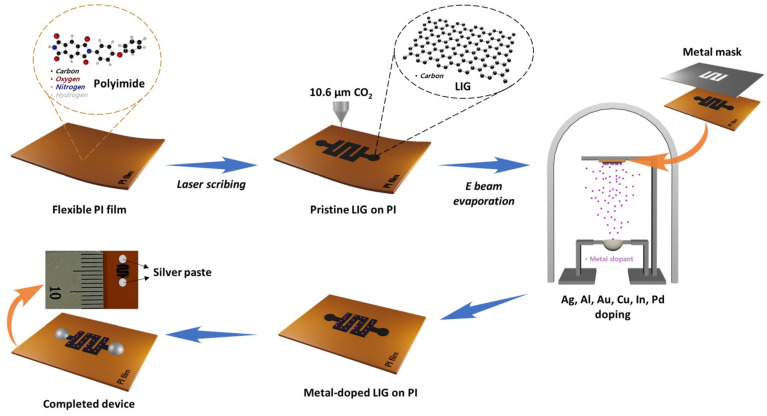
Schematics of the fabrication of the P-LIG and M-LIG sensor devices.

## Data Availability

Data is contained within the article and Appendix A.

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
