# Peer review of "Investigation of Laser-Induced Graphene (LIG) on a Flexible Substrate and Its Functionalization by Metal Doping for Gas-Sensing Applications"

_ijms, 2024, doi:10.3390/ijms25021172_

Round 1

Reviewer 1 Report

Comments and Suggestions for Authors

Recommendation: The manuscript is acceptable 

There couple of points that authors could take care of those;

(1) There are some typographical error in the manuscript (like mobility unit), they should take care of those throughout the manuscript

(2) They can increase the resolution of the figures, current format it's very hard to read them. (like Figure 3, it's difficult to identify the colors).

(3) In XPS study, they have potted the survey reports, author should mention the other peak identifications. And what are the atomic percentages for each elements. They could consider that point.

(3) In the conclusion section they should mention the point about the future perspective about this study or sensing. What could be the specific applications of this study?

Comments on the Quality of English Language

 Minor editing of English language required

Reviewer 2 Report

Comments and Suggestions for Authors

The paper utilized the LDW method to create LIG gas-sensor devices on flexible PI films, investigating their sensing performance towards NO2 and NH3. Comparative analysis with various metals indicated that doping metals into graphene-enhanced response to NH3 but the diminished response to NO2, highlighting the potential for developing high-performance LIG-based gas sensors. The manuscript is well-organized and provides a comprehensive examination of the gas-sensing characteristics of laser-induced graphene (LIG). Here are some comments that may assist in enhancing this paper:

1-    I recommend moving the material section before the result section.

2-    It's good that you highlight the limitations of theoretical studies and the need for practical and experimental studies.

3-    When discussing metal doping, consider briefly explaining the role of the Dirac-cone band structure and zero bandgap in graphene.

4-    For the benefits of LDW, consider providing examples of how rapid production and precision patterning have been advantageous in practical applications.

5-    Enhance your introduction by incorporating practical applications of other decorated carbon materials, such as CNF (refer to: DOI: 10.3390/ma15041383).

6-    Why the heteroatom doping method was not suitable in this case?

7-    Does this oxidation have any intended or unintended effects on gas sensing?

8-    Briefly elaborate on the rationale behind choosing the laser parameters (power, speed, wavelength) and how they impact the LIG fabrication.

9-    Explain the significance of optimizing the operation temperature at 100°C for all tests. And how does further temperature increase affect the response? Discuss the relationship between temperature, Debye length, and charge-carrier density in the context of gas adsorption and desorption.

10- Discuss the observed saturation point for NO2 at 500 ppm and the lack of substantial variation for elevated NH3 concentrations.

11- The text in Figure 6 is not clear.

Comments on the Quality of English Language

The English writing in the manuscript is generally clear and understandable. The sentences are well-constructed, and the technical terminology is used appropriately. 
